# Tracing the Origin of the *RSPO2* Long-Hair Allele and Epistatic Interaction between *FGF5* and *RSPO2* in Sapsaree Dog

**DOI:** 10.3390/genes13010102

**Published:** 2022-01-01

**Authors:** Mingue Kang, Byeongyong Ahn, Seungyeon Youk, Yun-Mi Lee, Jong-Joo Kim, Ji-Hong Ha, Chankyu Park

**Affiliations:** 1Department of Stem Cell and Regenerative Biotechnology, Konkuk University, Seoul 05029, Korea; mingue5349@gmail.com (M.K.); ahn.b@outlook.com (B.A.); dbrtmddus@hanmail.net (S.Y.); 2Department of Biotechnology, Yeungnam University, Gyeongsan 36461, Korea; ymlee@yu.ac.kr (Y.-M.L.); kimjj@ynu.ac.kr (J.-J.K.); 3Korean Sapsaree Foundation, Gyeongsan 38412, Korea; jhha@knu.ac.kr

**Keywords:** Sapsaree dogs, hair length, *RSPO2*, *FGF5*, Korean native dogs

## Abstract

Genetic analysis of the hair-length of Sapsaree dogs, a Korean native dog breed, showed a dominant mode of inheritance for long hair. Genome-Wide Association Study (GWAS) analysis and subsequent Mendelian segregation analysis revealed an association between *OXR1*, *RSPO2*, and *PKHD1L1* on chromosome 13 (CFA13). We identified the previously reported 167 bp insertion in *RSPO2* 3’ untranslated region as a causative mutation for hair length variations. The analysis of 118 dog breeds and wolves revealed the selection signature on CFA13 in long-haired breeds. Haplotype analysis showed the association of only a few specific haplotypes to the breeds carrying the 167 bp insertion. The genetic diversity in the neighboring region linked to the insertion was higher in Sapsarees than in other Asian and European dog breeds carrying the same variation, suggesting an older history of its insertion in the Sapsaree genome than in that of the other breeds analyzed in this study. Our results show that the *RSPO2* 3’ UTR insertion is responsible for not only the furnishing phenotype but also determining the hair length of the entire body depending on the genetic background, suggesting an epistatic interaction between *FGF5* and *RSPO2* influencing the hair-length phenotype in dogs.

## 1. Introduction

Mammalian hair length is regulated by the activation and inhibition of the Wnt signaling pathway in the hair dermal papilla (DP) and hair matrix, which determine the hair growth cycle divided into anagen, catagen, and telogen [1,2,3,4,5]. In previous studies, the R spondin family, including *R-spondin-2* (*RSPO2*), has been identified to maintain the anagen phase by forming a positive feedback loop of Wnt signaling by inhibiting the Znrf3/Rnf43 complex that antagonizes the Wnt signaling pathway in the hair matrix [5,6,7]. Additionally, *fibroblast-growth-factor-5* (*FGF5*) induces the catagen phase by terminating the positive feedback loop by suppressing R-spondin expression in DP through the Fgf signaling pathway [5,8,9]. 

The hair phenotype of modern domestic dog (*Canis lupus familiaris*) breeds is tremendously diverse and is an essential characteristic of each breed. Previous studies have identified mutations in three genes, namely *FGF5*, *RSPO2*, and *keratin-71* (*KRT71*), to be associated with hair length, furnishing, and curl shape, respectively, in dogs [10,11]. The mutations in *FGF5* and *KRT71* were nucleotide substitutions, but the *RSPO2* variant was caused by the 167 bp insertion of a 9-nucleotide (ATAATGAAC) repeat located in the 3’ UTR of *RSPO2* transcripts. 

Most domestic dog breeds have been developed in the last two hundred years as a result of direct selection designed to meet working or aesthetic requirements [12,13,14,15]. The Sapsaree is a medium-sized, long-haired breed of dog native to Korea and frequently appears in the paintings of the Chosun dynasty, the last imperial dynasty of Korea, indicating their presence in the Korean peninsula for at least a few hundred years [16]. Sapsarees were close to extinction in the mid-1980 but were restored from a small number of animals in Korea. During the restoration process, offspring with short-hair phenotypes different from those of the long-haired parents appeared, but the underlying mechanism for hair length variation has not been elucidated. The short-haired Sapsaree has also appeared in the old paintings of the Chosun dynasty (http://www.sapsaree.org/; accessed on 26 September 2021) [16].

In this study, we showed that the hair phenotypes of Sapsarees were caused by the same mutation previously identified in the *RSPO2* gene. We also analyzed selection signatures for the hair length phenotype and analyzed the phylogenetic relationships of the identified haplotypes for the region in domestic dogs. We also investigated the mode of inheritance of *FGF5* and *RSPO2* for hair length phenotypes in domestic dogs.

## 2. Materials and Methods

### 2.1. Animals and DNA Preparation

Blood samples and pedigree information of 61 long-haired and 53 short-haired Sapsaree dogs were provided by the Sapsaree Research Foundation in Gyeongsan, Korea. Genomic DNA was isolated from 1 mL of peripheral blood using the QIAamp DNA Blood Midi Kit (Qiagen, Hilden, Germany) according to the manufacturer’s protocol. Animal welfare and experimental procedures were approved by the Institutional Animal Care and Use Committee (IACUC) of Konkuk University and conducted according to the accepted protocol.

### 2.2. Genotyping

Single nucleotide polymorphism (SNP) typing was performed using an Illumina 170 K CanineHD BeadChip (Illumina, San Diego, CA, USA) according to the manufacturer’s protocols. Genotype calls were generated using an Illumina GenomeStudio Genotyping Module (Illumina). Pairs of individuals with genetic similarities above 99% were regarded as twins by the identity-by-state test implemented using PLINK v1.07 [17]. Low-quality SNPs with minor allele frequency (MAF) < 0.05% and genotype call rate < 90%, and those that significantly deviated from the Hardy–Weinberg equilibrium (*p* < 0.000001) were removed. 

### 2.3. GWAS and Linkage Analysis

The hair length phenotype (long versus short) data were modeled using a generalized linear mixed model. The logistic regression
(1)$$\log\left(\frac{\pi \left(x\right)}{1-\pi \left(x\right)}\right)=Xb+SNPp+Zu+e$$
was fitted: *b* vector containing the fixed effects associated with the traits, while gender was fitted as a fixed effect and age as a covariate; *X* is the design matrix containing the fixed effects related to the trait, *p* is the vector of the effect of SNP effect, *u* represents the vector of the random effects related to the animal effects for the trait, *Z* is the design matrix that corresponds to the animal effects of the trait, and *e* is a vector containing the residual. The analysis was conducted using the SAS software version 9.1 (www.sas.com; accessed on 26 September 2021). SNPs linked genomic regions were estimated using PLINK 1.9 (www.cog-genomics.org/plink/1.9/; accessed on 26 September 2021; [18]). The filtering criteria were the same as described above. Linkage disequilibrium (LD) window size was set as 1000 kb, and variants with LD (r^2^) > 0.7 were used for linkage analysis.

### 2.4. Whole-Genome Sequencing Data

The variant call format (VCF) data of 722 dogs was retrieved from NCBI (accession PRJNA448733, https://www.ncbi.nlm.nih.gov/bioproject/; accessed on 26 September 2021). In addition, the genome sequencing data of 151 dogs, including five breeds of Northeast Asian breeds (PRJNA782070), were retrieved from the NCBI Short Read Archive database, and variant calling was conducted (Appendix A). Briefly, whole-genome sequencing data were mapped to the dog reference genome CanFam3.1 (RefSeq accession: GCF_000002285.3) using BWA MEM v0.7.17-r1188 [19]. Base qualities were recalibrated using BaseRecalibrator in the Genome Analysis Toolkit (GATK) package v4.1.9.0 [20]. The joint variant was created using GATK GenotypeGVCFs. Regions of high strand bias (Fisher Strand > 30.0), low quality (QualityOfDepth (QD) < 2), and high complexity with more than three SNPs within a 35 bp window were removed using GATK VariantFiltration. To create a merged VCF file, common SNPs from 151 VCF data, including 5 Sapsarees and 722 VCF data from Plassais et al. [15], were integrated using the -merge option of bcftools 1.13. SNPs were subsequently filtered according to the targeted genomic regions using the thresholds of MAF < 0.01, minimum filtered depth (DP) < 10, and minimum genotype quality (GQ) < 30. Subsequently, individuals with variant call rates under 90% were filtered out. Data from the mixed or unknown breeds were also removed. The same criteria were used for all analyses based on the merged VCF file.

### 2.5. PCR and Sanger Sequencing

Genotyping PCR was conducted using 15 ng of DNA to determine the presence and absence of the 9-nucleotide repeat insertion in the 3’ UTR of *RSPO2* using previously described primers (forward: 5′-TGGCTAAAGAAAACTTCCACAA-3′; reverse: 5′-AAATTACCATCATGAGACCATGC-3′) [11]. PCR products were analyzed on a 1% agarose gel, and the genotype was determined according to the product sizes (817 and 676 bp for insertion and wildtype alleles, respectively). PCR products were prepared for direct sequencing as described previously [21]. Sequencing reactions were carried out with 30 ng of amplicons from homozygous individuals and 2 pmol of primers identical to PCR amplification for both directions using an ABI PRISM BigDye^TM^ Terminator Cycle Sequencing Kit (Applied Biosystems, Massachusetts, USA) following the manufacturer’s protocol. The reactions were analyzed using an ABI 3730 Genetic Analyzer (Applied Biosystems, Waltham, MA, USA). The sequencing chromatograms were analyzed using CLC main workbench 7.8.1 software (CLC bio, Aarhus, Denmark).

### 2.6. Tajima’s D and Nucleotide Diversity

Tajima’s D was calculated using the Tajima’s D option in vcftools 0.1.17 [22], with a window size of 10,000. Pairwise nucleotide diversity (π) of a genomic region was calculated for breeds with typing data of more than three individuals using vcftools 0.1.17 [22] with “--window-pi 20000,” and “--window-pi-step 10000” options. 

### 2.7. Haplotype Analysis

Haplotype blocks were determined using PLINK v1.9 by assigning the case (*RSPO2* 3’UTR insertion homozygotes) and control (*RSPO2* 3’UTR wildtype homozygotes) for each individual. The genotype information of *RSPO2* has been reported previously [11]. The output files were used to define haplotype blocks using Haploview 4.2 [23], and each block was defined by confidence intervals from Gabriel et al. [24] with default options. Haplotype phasing for the estimated haplotype blocks was conducted using BEAGLE v 5.2 [25] with default options. The breeds with SNP typing results from more than three animals were used. For the phased haplotype sequence, a haplotype tree was constructed using Haplotype-Viewer [26], and pairwise distances among haplotypes were estimated using Dnaml [27]. The geographical origin of dog breeds was referred from the Federation Cynologique Internationale (FCI. http://www.fci.be/en/; accessed on 26 September 2021).

### 2.8. Phylogenetic Analysis and Fixation Index (F_ST_^R^)

Unweighted Pair Group Method with Arithmetic Mean (UPGMA) clustering and neighbor-joining trees were constructed using VCF2poptree [28] with a pairwise diversity output format and visualized using MEGA-X 10.2.2 [29]. Evolutionary distances between individuals were estimated based on the difference in the number of SNPs. F_ST_^R^ was calculated using a published R script [30] using a variant call rate of 0.95, according to the following equation [31]: (2)$$F_{ST}^{\;\;\;\;R}=\frac{\sum_{k}{\hat{N}}^{\left[k\right]}}{\sum_{k}{\hat{D}}^{\left[k\right]}}$$

For F_ST_^R^ estimation, dog breeds with data of more than four individuals were selected. 

### 2.9. Multidimensional Scaling

Multidimensional scaling (MDS) was performed using the informative SNPs in the CFA13:8,530,000-8,730,000 bp region from 363 dogs. A pairwise genetic distance matrix was calculated according to the ratio of the nucleotide difference (p-distance) between individuals using VCF2Dis (https://github.com/BGI-shenzhen/VCF2Dis; accessed on 26 September 2021), and the result was displayed in the MDS plot using the cmdscale function of R (ver 4.1.1; R Core Team, 2021).

## 3. Results

### 3.1. Dominant Inheritance of the Long-Hair Phenotype in Sapsarees 

The Sapsaree is a Korean native dog breed with medium body size; long hair is its most prominent feature [32]. However, individuals with short hair also appear in the population (Appendix A). To understand the mode of inheritance and the variation in the fur phenotype of Sapsarees, we evaluated the Mendelian segregation of the fur phenotypes of 114 Sapsarees, 61 long-haired and 53 short-haired individuals (Appendix A). The pedigree information of the dogs is described in Appendix A. In the population, we identified two crosses with both long-haired parents that produced both long- and short-haired offspring in the same litter. The total number of long- and short-haired offspring from the crosses was two and three, respectively. In contrast, crosses between short-haired parents produced offspring with a phenotype identical to that of the parents in three crosses. The results indicate the dominant Mendelian inheritance of long-hair over the short-hair phenotype, and there was no exception.

### 3.2. Mapping the Sapsaree Hair Length Variation-Associated Region to Dog Chromosome 13 

Genome-wide SNP typing of 114 Sapsarees using the 170 K canine chip was carried out for a GWAS analysis to identify hair length phenotype-associated regions in the Sapsaree genome. Low-quality SNPs were filtered out, and logistic regression analysis was performed using GLMM. We found that 18 out of 79,229 informative SNPs showed significant associations (*p*-value < 10^−5^) with the hair length variation of Sapsarees (Appendix A). The 18 SNPs were mapped to seven different *Canis familiaris* chromosomes (CFA) with the greatest number of SNPs (*n* = 8) on CFA 13. However, none of the significant SNPs were found in the coding region.

### 3.3. Identification of RSPO2 as a Candidate for the Hair Length Variation of Sapsarees 

The annotation of the eight SNPs on CFA13 showed that one SNP was located in the non-coding RNA (ncRNA) (LOC111098552), two SNPs in the intronic region of *CSMD3*, and the rest in the intergenic region (Figure 1). However, none of the genes near the SNPs were identified as putative candidate genes for the hair phenotype, considering their ontology. Therefore, we manually conducted Mendelian segregation analysis for the genotyping data of CFA 13 using a 170 K canine chip and identified an additional nine SNPs unidentified in the GWAS analysis for their associations with the hair phenotypes (Appendix A). We then conducted a linkage analysis for all the identified SNPs (*n* = 17) to further define the regions of candidate genes (r^2^ > 0.7) using PLINK against the VCF of 151 dogs of 20 selected breeds according to geographical origins and hair phenotypes (Appendix A: source 1). We found that the regions containing *OXR1*, *RSPO2*, and *PKHD1L1* were strongly associated with the SNPs (Appendix A). Interestingly, this result corresponds to previous studies in which the upregulation of *RSPO2* expression by the insertion of a 167 bp segment of a nine-nucleotide repeat (namely, *RSPO2*-repeat) in the 3′ untranslated region (UTR) caused the furnishing phenotype of dogs [11]. Therefore, we assessed the presence of the *RSPO2*-repeat in long-haired Sapsarees by sequence analysis of the *RSPO2* 3′ UTR-specific amplicons (Appendix A) and found *RSPO2*-repeat (nine repeats of ATAATGAAC) in only long-haired Sapsarees. 

### 3.4. Confirmation of the RSPO2-Repeat Insertion as the Underlying Mechanism of Hair Phenotype Variations in Sapsarees using Pedigree Typing 

Because the *RSPO2*-repeat was reported as the causative mutation for the furnishing phenotype in dogs, we investigated if the *RSPO2*-repeat was responsible for the phenotype of long-haired Sapsarees using *RSPO2*-repeat typing against the Sapsaree pedigree. The genotyping PCR amplification produced 817 or 676 bp amplicons depending on the presence or absence of the *RSPO2*-repeat, respectively. We confirmed the complete correlation between the long-hair phenotype and the presence of 817 bp products (Figure 2). The 676 bp allele was detected in both long and short-haired Sapsarees, which was consistent with the dominant inheritance of the long-hair allele in Sapsarees. These results indicate that the *RSPO2*-repeat was indeed responsible for the long-hair phenotype in Sapsarees.

### 3.5. Selective Sweep of the RSPO2-Associated Regions in Long-Haired Dog Breeds

To evaluate the presence of selection signatures of the *RSPO2* region on CFA 13 (CFA13:8,110,227–9,111,951 bp) among long-haired dog breeds, we analyzed the allele frequency and distribution pattern of 1,213 SNPs identified from a 1 Mb region on CFA 13—500 kb upstream and downstream each of *RSPO2* 3′ UTR—using two different datasets, that is, 18 breeds of 56 *RSPO2*-repeat homozygous individuals and 118 breeds of 369 dogs regardless of *RSPO2* genotype (Appendix A). The results showed the complete fixation of an allele in CFA13:8,300,000–8,310,000, 8,500,000–8,510,000, 8,620,000–8,630,000, 8,660,000–8,680,000, and 8,950,000–8,960,000, 8,970,000-8,980,000 bp regions in *RSPO2*-repeat homozygotes (Appendix A). Selective sweeps with Tajima’s D > 95% confidence interval (CI) were identified from the regions, CFA13:8,300,000-8,310,000, 8,500,000–8,700,000, and 8,950,000–9,060,000 bp (Figure 3). The *RSPO2* 3′ UTR was also included in the region of selection sweep. In contrast, Tajima’s D values from the dataset, including both long- and short-haired dog breeds, did not display a selection signature for the region. 

### 3.6. Common Origin of the RSPO2 Allele among Diverse Long-Hair Dog Breeds 

To understand the ancestry of the *RSPO2*-repeat allele in long-haired dog breeds, phylogenetic analysis was carried out using 770 SNPs identified from a 200 kb region surrounding the *RSPO2* 3′ UTR (CFA13:8,530,000−8,730,000 bp) from 92 individuals of 24 breeds selected based on their geological origins (Figure 4, Appendix A, and Appendix A). In both neighbor-joining and UPGMA trees, the individuals of long-haired breeds, including the Shih Tzu (SIT), Tibetan Terrier (TIT), Portuguese Water Dog (PWD), Yorkshire Terrier (YOT), and long-haired Sapsarees (SAP), were clustered close (Figure 4 and Appendix A). Chinese crested dogs (CHC) were also clustered within the long-haired dog cluster, suggesting the presence of the *RSPO2*-repeat in the breed and additional changes in hair phenotypes due to the influence of other mutations on their hairless body with a crested hair phenotype. The average genetic distance among the long-haired Sapsarees (SAP1, 2, and 3) based on the number of nucleotide differences (*n* = 22.67) was much larger than the average genetic distance between the other breeds in the cluster (*n* = 13.93), suggesting a possibly longer history of the long-hair phenotype of Sapsarees (Appendix A). Consistently, Sapsarees were shown as an outgroup within the long-haired dog cluster in the tree (Figure 4 and Appendix A). Homozygous short-haired Sapsarees (SAP4 and 5) formed clusters with short-haired breeds native to China and Korea, such as the Pug (PUG), Jindo (JIN), and DongGyeongi (DON) and other European breeds like the Golden Retriever (GOR), and Australian breed Australian Cattle dog (ACD). Gray wolf (GRW) and Himalayan wolf (*Canis lupus chanco*) (HIW) constituted a separate cluster. Phylogenetic relationships of the *RSPO2* region from short-haired dog breeds were more diverse and distantly related than those of the long-haired breeds. The MDS plot using the pairwise distance matrix generated from the CFA 13: 8,530,000−8,730,000 bp region, identical to the phylogenetic analysis from 363 dogs (Appendix A), also showed the clustering of the long-haired breeds with the *RSPO2*-repeat allele (Figure 5). These results suggest that the allelic origin of the *RSPO2* gene in most long-haired dog breeds is common.

### 3.7. Haplotype Diversity of the Region Linked to RSPO2-Repeat in Domestic Dogs 

The haplotype block associated with the *RSPO2*-repeat was estimated for the 200 kb region (CFA13:8,530,000−8,730,000 bp), which showed a selection signature. A total of 224 individuals−55 belonging to 19 breeds homozygous for *RSPO2*-repeat and 169 belonging to 35 breeds homozygous for *RSPO2*-wildtype allele−were used to define the haplotype blocks associated with the *RSPO2*-repeat allele (Appendix A). Among the 29 haplotype blocks defined, haplotype block 20 was the largest and consisted of 57 SNPs (Appendix A). For haplotype block 20, we identified 12 major haplotypes with a haplotype frequency >1% (Appendix A) that could be differentiated using 11 tag SNPs. Interestingly, a haplotype consisting of the SNPsT/G/T/A/G/T/T/A/A/G/A/A/G/C/T/T/G/C/A/A/A/G/T/G/A/G/G/G/A/A/T/G/T/C/T/A/C/G/T/A/T/G/G/A/A/A/C/T/C/G/C/T/G/C/A/T/C was highly specific to the breeds carrying the *RSPO2*-repeat allele, with a few exceptions (*p*-value = 7.13 × 10^−89^, Appendix A). Subsequently, an additional haplotype analysis using the entire dataset of 363 individuals including heterozygotes (*n* = 13), genotype-undefined individuals (*n* = 126), and homozygotes (*n* = 224), was conducted using the 57 SNPs, and the 12 major haplotypes (HapA to L) were defined (Figure 6). HapA was distributed across the largest number of breeds (*n* = 60). Hap J was the most distant from the other haplotypes. HapD was associated with the *RSPO2*-repeat insertion. The distribution of haplotypes in block 20 did not have any geological associations.

Because the complete differentiation of the *RSPO2-*repeat allele from the wildtype allele was not achieved by haplotype analysis using 57 SNPs, we conducted an additional haplotype phasing using all SNPs (*n* = 178) present in the haplotype block 20, which increased the number of haplotypes to 93 (Appendix A). Finally, five haplotypes were identified to be specific to the *RSPO2*-repeat allele-carrying dog breeds. In addition, five haplotypes (Hap1 to 5) shared two common SNPs at positions 8,610,423 (T) and 8,610,424 (A), in contrast to G and C, respectively, of the remaining haplotypes corresponding to short-haired breeds. Consequently, the occurrence of 93 haplotypes divided the animals into 278 *RSPO2*-wildtype homozygotes, 77 *RSPO2*-repeat homozygotes, and 8 *RSPO2* heterozygotes (Appendix A) and showed complete agreement with the reported genotypes for each breed [11]. Therefore, we suggest that the two SNPs can be used to predict the allelic type of *RSPO2* without the *RSPO2* 3′ UTR region. The five haplotypes associated with the *RSPO2*-repeat were closely clustered in the phylogenetic tree (Appendix A), which was in line with the clustering pattern in the MDS plot (Figure 5).

### 3.8. Genetic Diversity in the RSPO2-Linked Region in Sapsarees

We compared nucleotide diversity (π) of the genomic region linked to *RSPO2* (CFA13:8,530,000–8,730,000 bp) in 57 *RSPO2*-repeat homozygotes belonging to 13 breeds with at least three animals per breed (Appendix A). The mean π in a 20 kb window was estimated to be 0.000034 (SD = 0.000054). Interestingly, long-haired Sapsarees showed the most nucleotide diversity (π = 0.00018) (Appendix A), which is consistent with the results of the phylogenetic analysis in Figure 4. We also analyzed breed differentiation by estimating F_ST_^R^ for the CFA13:8,530,000–8,730,000 bp region between the breeds, each with at least four individuals (Appendix A). The pairwise F_ST_^R^ for the 200 kb of the *RSPO2*-linked region between Sapsarees and other breeds was lowest with Jack Russel Terrier (JRT) and highest with German Shepherd Dog (GSD). Northeast Asian dog breeds with relatively low F_ST_^R^ with Sapsarees were Jindo, DongGyeongi, and Shiba Inu (Figure 7), consistent with their geographical origins.

## 4. Discussion

Native breeds of domestic animals are valuable in helping unravel the molecular mechanisms underlying their diverse phenotypic characteristics obtained through their adaptation to natural or human-induced change. Several GWAS studies have identified genes or loci significantly associated with diverse morphological traits in dogs and highlighted the molecular mechanisms and importance of genetic variations in phenotypic diversity [15,33,34]. In this study, we determined the causative gene for hair length variations in Sapsarees, a native dog breed in the Korean peninsula, and investigated the allelic origin, selection signature, and haplotypic variations of the neighboring sequence of *RSPO2* across diverse breeds (*n* = 172) of domestic dogs (Appendix A). 

Our initial GWAS analysis mapped the region of association to the long-hair phenotype of Sapsarees on CFA 13, but with less precision. This may be due to the dominance of the long-hair allele and the small sample size (*n* = 114). Probing several other chromosomal regions showing significance also did not lead to the identification of any hair growth-related genes. However, the presence of multiple markers on CFA13 showing strong associations with the long-hair phenotype was critical for the identification of the causal variants underlying hair length variation in Sapsarees.

Sapsarees, a native dog breed, have existed in the Korean peninsula for at least a few hundred years and carry an identical allele responsible for the hair phenotype of the European dog breeds [35] despite distant genetic relationships between Sapsarees and Western dog breeds [36,37,38]. Currently, 197 long-haired dog breeds have been officially recognized by the American Kennel Club (AKC, www.ack.org/dog-breeds; accessed on 26 September 2021), and their hair length variations have been explained by the *FGF5* and *RSPO2* variations [11,39]. The presence of the identical *RSPO2*-repeat allele among the *RSPO2*-associated long-haired dog breeds, including Sapsarees, suggests a common origin of the allele or an ancestral effect in long-haired dog breeds. This suggests that the genetic origin of hair length phenotypes could be different from the genetic background estimated from whole genomes [13,40,41,42,43,44].

The presence of both *RSPO2*-repeat and wildtype alleles in a single breed has been reported on several occasions, including in the Portuguese Water Dog, Brussels Griffon, Dachshund, and Sapsarees [11,35]. The maintenance of other phenotypic characteristics, except for hair length in a breed, could be explained by the occurrence of genetic mutations in hair length association genes. However, it is less likely to carry the same causative allele across several breeds through independent mutational events. For example, five recessive variants of *FGF5* have been reported as causal mutations for the long-hair phenotype in diverse dog breeds with the *FGF5*:p.Cys95Phe mutation as the most common among the breeds [39]. In contrast, an identical *RSPO2* variant has been found in all the breeds analyzed to date, however, the repeat number of the nine-nucleotide repeat in the *RSPO2* 3′ UTR insertion allele differs slightly in some breeds [35]. Perhaps missense mutations in *RSPO2* could be harmful, considering its role in Wnt signaling, which is critical for animal development [45,46,47]. 

To understand the possible mechanisms involved in the insertion of the nine-nucleotide repeat in *RSPO2* 3′ UTR, we blasted 150 bp each of the upstream and downstream sequence of the 3′ insertion junction of *RSPO2* as a query against the reference genome of dogs using NCBI nucleotide BLAST (https://blast.ncbi.nlm.nih.gov/Blast.cgi; accessed on 26 September 2021). Although the results showed several matches on chromosomes other than CFA 13, we were unable to draw any conclusion on the mechanism of simple repeat insertion.

A previous study reported an *RSPO2*-associated region showing a high F_ST_ [48]. However, the region is less directly related to the 200 kb genomic region strongly linked to the *RSPO2*-repeat allele in Sapsarees. The higher genetic diversity in the 200 kb genomic region linked to the *RSPO2*-repeat allele in Sapsarees than the other dog breeds suggests that the age of the *RSPO2* 3′ UTR insertion in Sapsarees may be older or less affected by intensive selection than in the other breeds analyzed in this study despite the bottleneck effect of breed rebuilding. However, the haplotype diversity in the neighboring region of the *RSPO2*-wildtype allele in 119 dog breeds showed much more diversity (Hap6–93) than that in the neighboring region of the *RSPO2*-repeat allele (Hap1–5) (Appendix A). Unexpectedly, the degree of population differentiation (F_ST_^R^) against the sequences of the 200 kb region of Sapsarees was the lowest (0.0594) with Jack Russel Terrier, a European breed, followed by other Korean native breeds, namely Jindo and DongGyeongi (Figure 7). This may suggest a common origin of the *RSPO2*-repeat allele in the domestic dog breeds. 

In previous studies, the *RSPO2*-repeat allele has been described as a causal mutation for the furnishing phenotype. However, the combined genotype of *FGF5* and *RSPO2*, in which the presence and absence of mutant alleles were described as “+” and “-”, respectively, showed that the hair length of Sapsarees was also significantly affected by the *RSPO2*-repeat allele (Table 1). Among Northeast Asian dogs, the long-haired Sapsarees (*FGF5* +, *RSPO2* +) showed allelic combinations of the two genes identical to those of Shih Tzu and Tibetan Terrier. This pattern also appeared in European breeds such as the Old English Sheepdog, Maltese, and Portuguese Water Dog [11]. In contrast, short-haired Sapsarees had the same allelic combination as that of other Northeast Asian breeds, including Pekingese and Tibetan Mastiff, as well as European breeds such as the Newfoundland, Golden Retriever, and English Cocker Spaniel [11]. However, none of the breeds with *FGF5* (−) and *RSPO2* (+) genotype showed the long-hair phenotype. The hair length of the short-haired Sapsarees (*FGF5* +, *RSPO2* −) was significantly shorter than that of other breeds with the same genotype, such as Pekingese and Tibetan Mastiffs. Such phenotypic variation could be due to the interaction of *RSPO2* with other hair cycle regulatory genes or the regulatory sequence variation of *RSPO2*. These results suggest that hair length in dogs is regulated by the epistatic interaction between *FGF5* and *RSPO2*. 

In this study, we identified *RSPO2* as the causative mutation determining the short hair length phenotype of Sapsarees and provided evidence for the role of the epistatic interaction between *RSPO2* and *FGF5* in hair length phenotype in domestic dogs. We also showed the presence of the *RSPO2* variant-associated selection sweep and haplotypic diversity of the region. The presence of a single *RSPO2* variant in all dog breeds strongly suggests a common origin of the mutation. The significantly higher genetic diversity within the *RSPO2*-repeat allele-linked haplotype block in Sapsarees than in the other long-haired dog breeds may suggest that the integration of the *RSPO2* variant in the genome of Sapsarees happened earlier than in that of other long-hair dog breeds carrying the variant. However, the difference in breeding history, including the intensity of inbreeding, should also be considered. Previous studies have suggested the *RSPO2*-repeat allele to be responsible for the long-hair phenotype in various dog breeds. However, this study further explored the phylogenetics of the *RSPO2* mutation in domestic dogs and its genetic interactions by also analyzing the genetic data of previously undescribed breeds.

## Figures and Tables

**Figure 1 genes-13-00102-f001:**
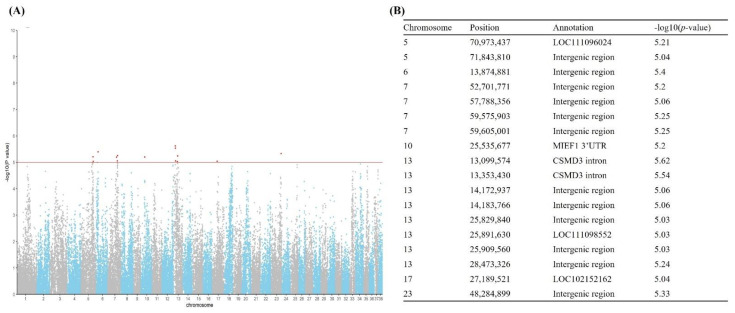
Hair length phenotype analysis using 114 Sapsarees genotyped on 170 K SNP makers. (**A**) Manhattan plot; (**B**) Maker list showing significance (*p*-value < 10^−5^).

**Figure 2 genes-13-00102-f002:**
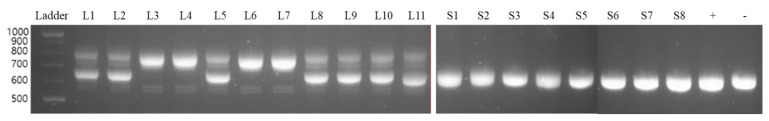
Gel image of the genotyping PCR of *RSPO2* 3’ UTR to differentiate the long- and short-hair allele. Animal ID is shown on top with their phenotypes (L and S for long- and short-hair phenotypes, respectively). The target size of the PCR product for the long- and short-hair alleles was 817 bp and 676 bp, respectively. “+” and “-” indicates positive and negative controls, respectively.

**Figure 3 genes-13-00102-f003:**
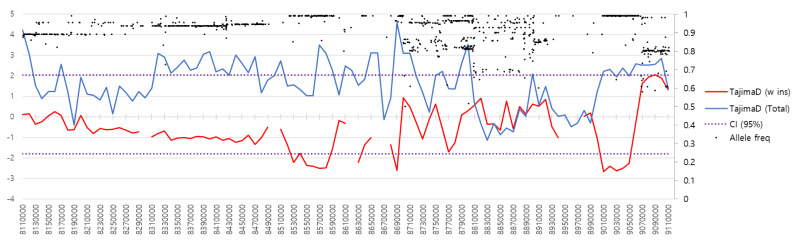
Allele frequency and selection signature for the *RSPO2* 3’ UTR-linked genomic region on CFA 13. The allele frequencies of 1,225 SNPs in the CFA13:8,110,000−9,110,000 bp region of 56 *RSPO2* 3’ UTR insertion homozygous dogs are indicated by black dots. Tajima’s D is depicted as red and blue lines for 56 dogs homozygous for the *RSPO2* 3′ UTR insertion and 369 dogs consisting of all genotypes. Dotted lines indicate the 95% confidence interval (CI) of Tajima’s D value according to the beta distribution corresponding to *n* = 55.

**Figure 4 genes-13-00102-f004:**
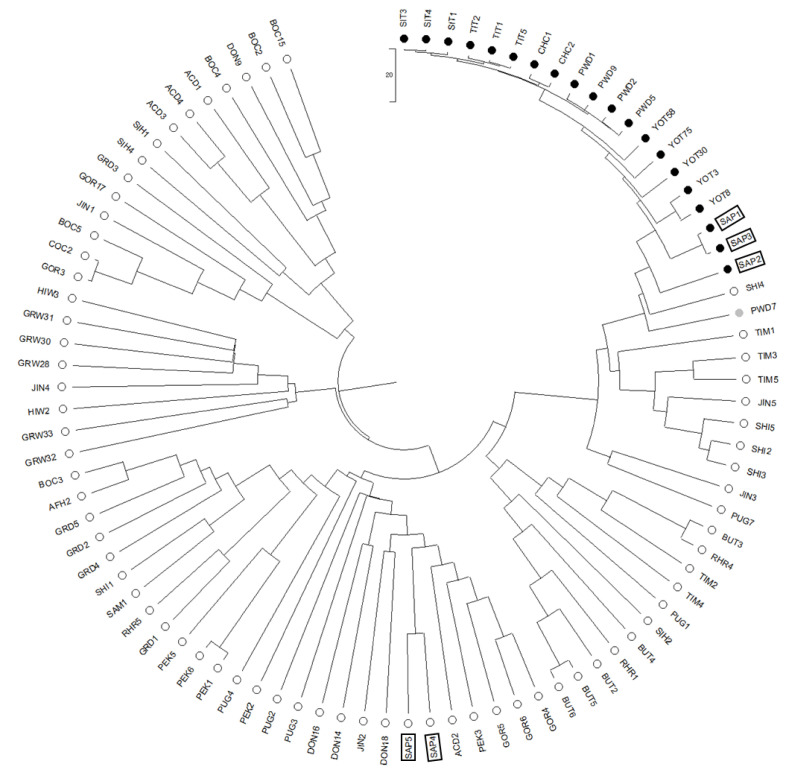
A phylogenetic tree for the 200 kb region on dog chromosome 13 with selection sweep for the hair length phenotype. A UPGMA phylogenetic tree was constructed using 785 SNPs from a total of 92 dogs belonging to 24 breeds with various hair length phenotypes. The branch length was determined on the basis of pairwise nucleotide differences in the CFA13:8,530,000−8,730,000 bp region among individuals. Homozygotes for the *RSPO2* 3’ UTR insertion are indicated by closed circles, and animals without the insertion are indicated by open circles. Gray circles indicate heterozygotes. Five Sapsarees were additionally marked with black squares. GOR (Golden Retriever), ACD (Australian Cattle Dog), AFH (Afghan Hound), YOT (Yorkshire Terrier), BOC (Border Collie), BUT (Bull Terrier), GRD (Great Dane), CHC (Chinese Crested), COC (Chow Chow), PWD (Portuguese Water Dog), RHR (Rhodesian Ridgeback), SIH (Siberian Husky), SAM (Samoyed), DON (DongGyeongi), GRW (Grey Wolf), HIW (Himalayan Wolf), JIN (Jindo), PEK (Pekingese), PUG (Pug), SAP (Sapsaree), SHI (Shiba Inu), SIT (Shih Tzu), TIM (Tibetan Mastiff), TIT (Tibetan Terrier).

**Figure 5 genes-13-00102-f005:**
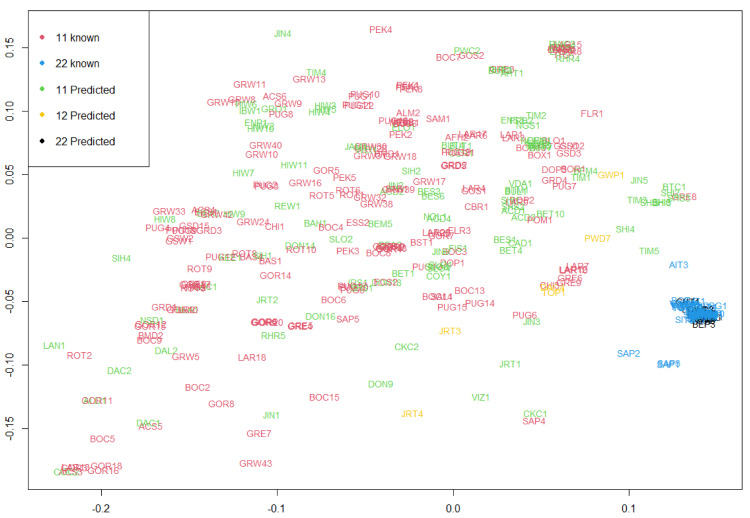
Multidimensional scaling plot of dog breeds with various hair lengths according to SNP variations. The plot was created on the basis of pairwise distances calculated from SNPs of the haplotype block 20 (CFA13:8,530,000−8,730,000 bp) linked to the *RSPO2* 3’ UTR region. The genotypes of *RSPO2* 3’ UTR in the top left of the plot were described as follows: “11” for homozygotes of the *RSPO2*-wildtype allele, “22” for homozygotes of the *RSPO2*-repeat allele, and “12” for heterozygotes. The words “known” and “predicted” indicate the use of available data (Cadieu et al. 2009) and prediction using the linkage analysis in this study, respectively. For information on the abbreviation of each breed, refer to Appendix A.

**Figure 6 genes-13-00102-f006:**
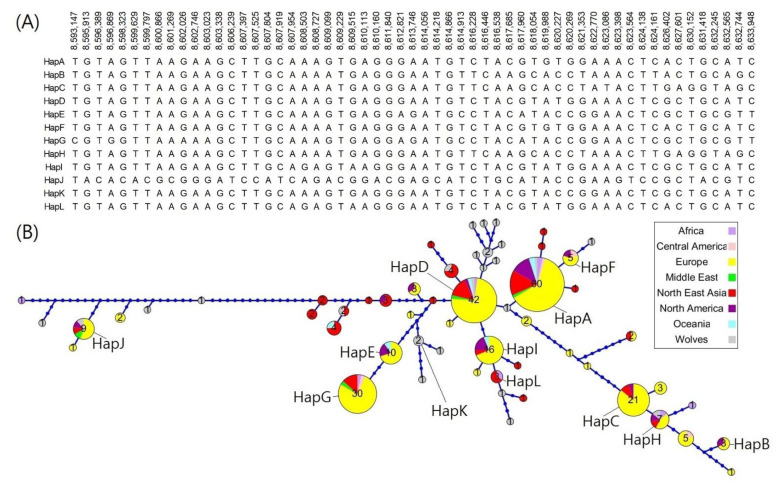
Haplotype tree showing the relationships of the 58 haplotypes belonging to the haplotype block 20 of the *RSPO2* 3’ UTR-linked region on dog chromosome 13. The 12 major haplotypes (HapA to L) with frequencies >0.01 are shown with the corresponding SNPs and chromosomal coordinates in (**A**) and indicated within or next to each haplotype node in the haplotype tree (**B**). Haplotype nodes are colored differently according to the geological origin of the breed, and the number of breeds with the corresponding haplotype is specified at each node. The distance between the nodes was determined based on nucleotide differences among haplotype sequences, and each node represents a difference of 1 SNP from the adjacent nodes or dots. Blue dots indicate intermediate sequence states between two different haplotype node sequences.

**Figure 7 genes-13-00102-f007:**
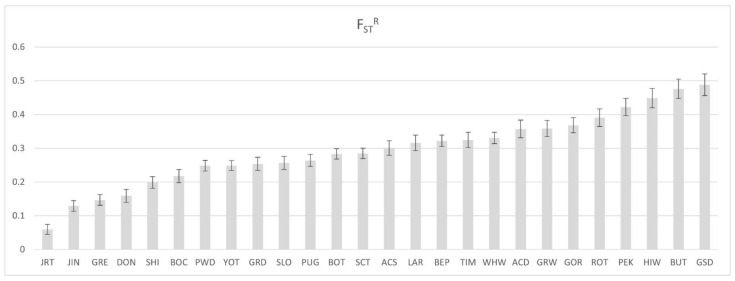
Pairwise F_ST_^R^ calculated from the *RSPO2* 3’ UTR-associated region of the long- and short-haired Sapsarees and other dog breeds. The pairwise F_ST_^R^ between the breeds was calculated based on the definition of Reich et al. (2009). The error bar at the top of each bar graph indicates the 95% confidence interval.

**Table 1 genes-13-00102-t001:** Hair length phenotypes and the presence of *FGF5* and *RSPO2* variants in Northeast Asian breeds and two selected European breeds.

Breed	Origin	Hair length	Furnishing	*FGF5*	*RSPO2*
DongGyeongi	Northeast Asia	Short	No	−	−
Jindo	Northeast Asia	Short	No	−	−
Pekingese	Northeast Asia	Long	No	+	−
Pug	Northeast Asia	Short	No	−	−
Sapsaree-long	Northeast Asia	Long	Yes	+	+
Sapsaree-short	Northeast Asia	Short	No	+	-
Shiba Inu	Northeast Asia	Short	No	−	−
Shih Tzu	Northeast Asia	Long	Yes	+	+
Tibetan Mastiff	Northeast Asia	Medium	No	+	−
Tibetan Terrier	Northeast Asia	Long	Yes	+	+
Petit Basset Griffon Vendeen*	Europe	Short	Yes	−	+
Irish Wolfhound*	Europe	Short	Yes	−	+

Note: Plus (+) and minus signs (−) indicate the presence and absence of the long-hair variant allele, respectively. The allele information and hair phenotypes of breeds marked with “*” is from Cadieu et al. (2009).

## Data Availability

All genetic data used in this paper are accessible through the registered accession number, and all other information is specified in the paper.

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
