# Peer review of "Tracing the Origin of the RSPO2 Long-Hair Allele and Epistatic Interaction between FGF5 and RSPO2 in Sapsaree Dog"

_genes, 2022, doi:10.3390/genes13010102_

Round 1
Reviewer 1 Report
Line 214 – It would be appropriate to add ´if´ after the word ´investigated´ - ´we investigated if the RSPO2-repeat is responsible…´
Figure 5 – this figure is confusing and should be rearranged and amended to make the graph easier to understand. The similarity of the colours used for ´known´ and ´predicted´ makes it hard to differentiate. Different colours should be used for ´known´ and ´predicted´ values.
Line 299 – formatting error, separated words ´consisting of the SNPs…´
Line 514, 539, 585 – the date of publication (year) should be written in bold letters, same as in all the other references.
Overall the study is done exceptionally well, methods are replicable and the contribution is of high value.
Author Response
We revised the manuscript according to the four comments from the reviewer-1. The changes were described below and highlighted in yellow in the attached revision together with other minor changes commented by reviewer-2. Authors sincerely thank the reviewer's effort to improve our manuscript.
1) Line 214 – It would be appropriate to add ´if´ after the word ´investigated´ - ´we investigated if the RSPO2-repeat is responsible…´ -> changed accordingly.
2) Figure 5 – this figure is confusing and should be rearranged and amended to make the graph easier to understand. The similarity of the colours used for ´known´ and ´predicted´ makes it hard to differentiate. Different colours should be used for ´known´ and ´predicted´ values. -> we changed colors and made it easy to distinguish the known and predicted.
3) Line 299 – formatting error, separated words ´consisting of the SNPs…´ -> changed accordingly.
4) Line 514, 539, 585 – the date of publication (year) should be written in bold letters, same as in all the other references. -> changed accordingly.

Reviewer 2 Report
Review of the manuscript:
Kang et al: Genetic analysis of hair length phenotype of Sapsaree dogs - tracing the origin of the RSPO2 long hair allele and epistatic interaction between FGF5 and RSPO2.
In this study, the authors have shown that the hair phenotypes of Sapsarees were caused by the same 56 mutation previously identified in the RSPO2 gene, and they have analyzed the selection signatures for the hair length phenotype and the phylogenetic relationships of the identified haplotypes.
This manuscript is well written, with solid analysis, and my only comment is the long title of the manuscript, which I recommend shortening, something like this:
Tracing the origin of the RSPO2 long hair allele and epistatic interaction between FGF5 and RSPO2 in Sapsaree dog.
Author Response
We revised the manuscript according to the four comments from the reviewer-2. The changes were described below and highlighted in yellow in the attached revision together with other minor changes commented by reviewer-1. Authors sincerely thank the reviewer's effort to improve our manuscript.
1) This manuscript is well written, with solid analysis, and my only comment is the long title of the manuscript, which I recommend shortening, something like this: Tracing the origin of the RSPO2 long hair allele and epistatic interaction between FGF5 and RSPO2 in Sapsaree dog. -> Authors shortened the title as suggested by reviewer 2.
